# Proliferative exhausted CD8⁺ T cells exacerbate long-lasting anti-tumor effects in human papillomavirus-positive head and neck squamous cell carcinoma

**Danni Cheng[1†], Ke Qiu[1†], Yufang Rao[1†], Minzi Mao[1†], Li Li[2], Yan Wang[3], Yao Song[1], Junren Chen[4], Xiaowei Yi[1], Xiuli Shao[1], Shao Hui Huang[5], Yi Zhang[3], Xuemei Chen[3], Sisi Wu[3], Shuaishuai Yu[3], Jun Liu[1], Haiyang Wang[1], Xingchen Peng[6], Daibo Li[1], Lin Yang[7], Li Chen[7], Zhiye Ying[4], Yongbo Zheng[1], Meijun Zheng[1], Binwu Ying[1,8], Xiaoxi Zeng[1,4], Wei Zhang[2], Wei Xu[9], Geoffrey Liu[10], Fei Chen[1*], Haopeng Yu[1,4*], Yu Zhao[1,4*], Jianjun Ren[1,4*]**

[1]Department of Oto-Rhino-Laryngology, and National Clinical Research Center for Geriatrics, West China Hospital, West China Medical School, Sichuan University, Chengdu, China; [2]Institute of Clinical Pathology, West China Hospital, Sichuan University, Chengdu, China; [3]Research Core Facility of West China Hospital, Sichuan University, Chengdu, China; [4]West China Biomedical Big Data Center, West China Hospital, Sichuan University, Chengdu, China; [5]Department of Radiation Oncology, Princess Margaret Cancer Centre, University of Toronto, Toronto, Canada; [6]Department of Biotherapy and National Clinical Research Center for Geriatrics, Cancer Center, West China Hospital, Sichuan University, Chengdu, China; [7]MinSheng Ear-Nose-Throat Hospital, Chengdu, China; [8]Department of Laboratory Medicine, West China Hospital, Sichuan University, Chengdu, China; [9]Department of Biostatistics, Princess Margaret Cancer Centre and Dalla Lana School of Public Health, Toronto, Canada; [10]Department of Medicine, Division of Medical Oncology and Hematology, Princess Margaret Cancer Center, University Health Network, University of Toronto, Toronto, Canada

***For correspondence:**
hxchenfei@163.com (FC);
yuhaopeng@wchscu.cn (HY);
yutzhao@VIP.163.com (YZ);
Jianjun.Ren@scu.edu.cn (JR)

†These authors contributed
equally to this work

**Competing interest:** The authors
declare that no competing
interests exist.

**Reviewing Editor:** Kellie N
Smith, The Johns Hopkins
University School of Medicine,
United States

**Abstract** The survival prognosis of human papillomavirus (HPV)-positive and HPV-negative head and neck squamous cell carcinoma (HNSCC) is largely different, and little is known about the anti-tumor mechanism of tumor-infiltrated exhausted CD8⁺ T cells (Tex) in HNSCC. We performed cell-level multi-omics sequencing on human HNSCC samples to decipher the multi-dimensional characteristics of Tex cells. A proliferative exhausted CD8⁺ T cell cluster (P-Tex) which was beneficial to survival outcomes of patients with HPV-positive HNSCC was identified. Interestingly, P-Tex cells expressed CDK4 genes as high as cancer cells, which could be simultaneously inhibited by CDK4 inhibitors and might be a potential reason for the ineffectiveness of CDK4 inhibitors in treating HPV-positive HNSCC. P-Tex cells could aggregate in the antigen-presenting cell niches and activate certain signaling pathways. Together, our findings suggest a promising role for P-Tex cells in the prognosis of patients with HPV-positive HNSCC by providing modest but persistent anti-tumor effects.

## Editor's evaluation

This study provides fundamental insight into the functional impact of CDK4 inhibition on cells in the tumor microenvironment, which is of high importance and interest to the field. The compelling conclusion that proliferative exhausted T cells are associated with response in HPV+ head and neck cancer is supported by the cohort of 14 patients with paired tumor and adjacent normal tissue and rigorous bioinformatic analysis of nearly 50,000 single CD3+ T cell transcriptomes. This work will be of interest to researchers across tumor types and in other immunological fields of study.

## Introduction

The incidence of head and neck squamous cell carcinoma (HNSCC) has continued to rise and an annual increase of 1.08 million new cases is estimated by 2030, which is greatly attributed to the increasing rates of human papillomavirus (HPV) infection (*Bray et al., 2018*; *Ferlay et al., 2019*; *Näsman et al., 2020*). HPV-positive HNSCC and HPV-negative HNSCC displayed markedly different characteristics from pathogenesis to treatment outcomes and are visualized as two distinct clinical entities (*Ang et al., 2010*; *Wansom et al., 2010*).

T cell exhaustion within the tumor microenvironment (TME) is a newly discovered phenomenon resulting from persistent antigen stimulation from chronic virus infection or tumors, in which tumor-infiltrated CD8$^+$ T cells experience gradual alterations in their functional capacity while highly expressing multiple inhibitory receptors, including PD1, TIM3, LAG3, CTLA4, and TIGIT (*Wherry and Kurachi, 2015*; *Blank et al., 2019*). Recent studies have revealed that heterogeneity is a hallmark of T cell exhaustion, and several distinct subsets of exhausted CD8$^+$ T cells (Tex) have been identified, each with unique gene signatures, functional characteristics, and epigenetic modifications (*Miller et al., 2019*; *Beltra et al., 2020*). Meanwhile, specific transitions among those subsets have also been illustrated under certain circumstances and might be associated with retained effector function and enforced tumor control (*Im et al., 2016*; *Kim et al., 2020*). However, how T cell exhaustion affects the prognosis of HNSCC patients and how it differs in HPV-positive and HPV-negative HNSCC remain to be further clarified.

Recently, there has been increasing evidence showing that HPV-positive HNSCC displays a T cell-inflamed phenotype distinct from its HPV-negative counterparts, indicating that HPV infection is associated with increased T cell infiltration and effector cell activation (*Gameiro et al., 2018*; *Wang et al., 2019*). Meanwhile, HPV infection has also been proven to be associated with T cell exhaustion, in which HPV-positive HNSCC expressed higher levels of multiple T cell exhaustion markers such as PD1, TIM3, LAG3, and TIGIT compared to HPV-negative HNSCC, suggesting of stronger antigen-specific T cell immunity in HPV-positive HNSCC (*Gameiro et al., 2018*; *Kansy et al., 2017*). However, more sophisticated T cell landscapes related to the HPV status of HNSCC remain to be further clarified.

Cyclin-dependent kinase 4 (CDK4) inhibitors are introduced as novel drugs by targeting and disrupting the CDK4-related cell cycle progression of cancer cells in recent years (*Goel et al., 2018*; *Deng et al., 2018*). However, promising treatment outcomes of CDK4 inhibitors were only observed in HPV-negative HNSCC rather than HPV-positive HNSCC (*Adkins et al., 2019*; *Oppelt et al., 2021*; *van Caloen and Machiels, 2019*) due to the mutation differences of cell cycle-related genes in cancer cells, and less attention has been paid to the effects of CDK4 inhibitors on infiltrating T cells.

To address these questions, we applied cell-level multi-omics sequencing techniques to decipher the multi-dimensional characterization of tumor-infiltrating T cells and its association with overall survival (OS) in human HNSCC with different HPV status.

## Results

### P-Tex cells were identified in the transcriptomic landscape of T cells in HNSCC patients

To decipher the multi-dimensional characterization of tumor-infiltrating T cells in human HNSCC, we integrated multi-omics sequencing based on 5' droplet-based single-cell RNA (scRNA-seq), single-cell TCR sequencing (scTCR-seq), and spatial transcriptomics in HNSCC samples (mainly oropharyngeal cell carcinoma, the most representative type of HPV-related HNSCC), and further verified in vitro

(n=24, *Supplementary file 1* and *Figure 1a*). A total of 49,813 CD3$^+$ T cells in 14 paired HNSCC tumor and normal adjacent samples were obtained after quality control. And 11 T cell clusters with distinct gene signatures were defined (*Figure 1b*, *Supplementary file 2*), including four CD8$^+$ T cell clusters (C1, C4, C7, and C9), five CD4$^+$ T cell clusters (C2: naïve CD4$^+$ T cells, C3: regulatory T cells [Treg], C5: T helper cells, C8: CD4-Tex, and C10: T follicular helper cells), one γδ T cell cluster (C6), and one double negative T cell cluster (C0: DN, CD4$^-$CD8$^-$), with marker genes (*Brummelman et al., 2018*) shown in *Supplementary file 3*.

We specifically characterized marker genes of CD8$^+$ T cell into several panels based on their canonical biological function (proliferation, exhaustion, and cytotoxicity, shown in *Figure 1c*). Among the four CD8$^+$ T cell clusters, C1 was characterized by expressing multiple effector genes, including *NKG7*, *GZMH*, *IFNG*, and *KLRG1*, with a high cytotoxicity score but a low exhaustion score, thus representing effective CD8$^+$ T cell (Teff) (*Galletti et al., 2020*). C4 showed high expression of checkpoint marker genes, including *PDCD1*, *HAVCR2*, *LAG3*, and *TNFRSF9*, with both high cytotoxicity and exhaustion scores, which was consistent with the cell identity of terminally differentiated exhausted CD8$^+$ T cells (T-Tex) (*Wherry and Kurachi, 2015*). Notably, in addition to high expression of the above-mentioned checkpoint marker genes, C7 and C9 also displayed high expression levels of cell cycle-related genes, including *CCNA2*, *UBE2C*, and *CDK4*, as well as stem-like genes *MKI67* (marker gene of proliferation) and *ASPM* (involved in regulation of the mitotic spindle and coordination of mitotic processes), with high cytotoxicity, exhaustion, and proliferation genes, featuring a gene expression profile reminiscent of a previously reported Tex subset with high proliferative capacity (*Wagle et al., 2021*), which we defined as P-Tex in the present study.

Meanwhile, T cells appeared to exhibit distinct tissue distributions, with higher proportions of Treg, Teff, T-Tex, and P-Tex cells being observed in tumor tissues while more DN and CD4-Tex cells being observed in adjacent normal tissues (*Figure 1d*, *Figure 1—figure supplement 1a*). To evaluate individual heterogeneity, we further clustered the cells of each HNSCC sample and confirmed the existence of all cell clusters across all samples (*Figure 1—figure supplement 1b–c*).

Notably, the P-Tex cluster (~2819 cells) could be further partitioned into two subclusters, which we annotated as P-Tex1 and P-Tex2. More specifically, *CDK4* (a canonical cell cycle-related marker gene) was mainly expressed in P-Tex2, whereas *MKI67* (the most canonical marker gene for proliferation) was mainly expressed in P-Tex1 (*Figure 1e–f*), indicating that these two cell clusters might fulfill their proliferative capacity through different mechanisms. Moreover, to further investigate the potential upstream regulatory mechanisms in shaping the molecular characteristics of each unique T cell cluster, we analyzed the transcription factor networks that driving the expression of the top expression genes in each T cell cluster (shown in *Figure 1g* and *Supplementary file 4*). Specifically, MYBL2, BRCA1, E2F1, E2F8, EZH2, and TFDP1 were the identified upstream regulatory transcription factors that predominantly drove the expression of proliferation-related genes of P-Tex1 and P-Tex2.

Taken together, our clustering strategy generated 11 distinct T cell clusters in HNSCC, among which P-Tex cells with both exhausted and proliferative phenotypes were identified.

## Functional characteristics of P-Texs

To further investigate the functional characteristics of P-Tex cells, we characterized the function of marker genes by comparing with Teff and T-Tex clusters. Specifically, gene ontology (GO) enrichment analysis showed that T cell activation, lymphocyte differentiation, and viral gene expression were enriched in all three Tex cell clusters, whereas the regulation of the cell cycle, apoptosis, and certain immune responses were enriched in P-Texs, showing divergent functional specialization (*Figure 2a*). Meanwhile, the activation states of CD8$^+$ T cell subpopulations were quantitatively assessed based on the average expression of previously described activation-related gene signatures (*Azizi et al., 2018*) (also shown in *Supplementary file 3*). Our results showed that the T-Tex cluster was the most activated, followed by the two P-Tex clusters (*Figure 2b*, left). In addition, CD8$^+$ T cells in tumor tissues were more activated than those in adjacent normal tissues (*Figure 2b*, right top). And no significant difference in T cell activation states was observed between HPV-positive and HPV-negative samples (*Figure 2b*, right bottom).

Additionally, to further confirm that P-Texs displayed high cell cycle-related function, we performed gene set enrichment analysis (GSEA) using the gene set that represents the cell cycle pathway, and the results showed that two P-Tex clusters were more enriched in the cell cycle signal pathway than the

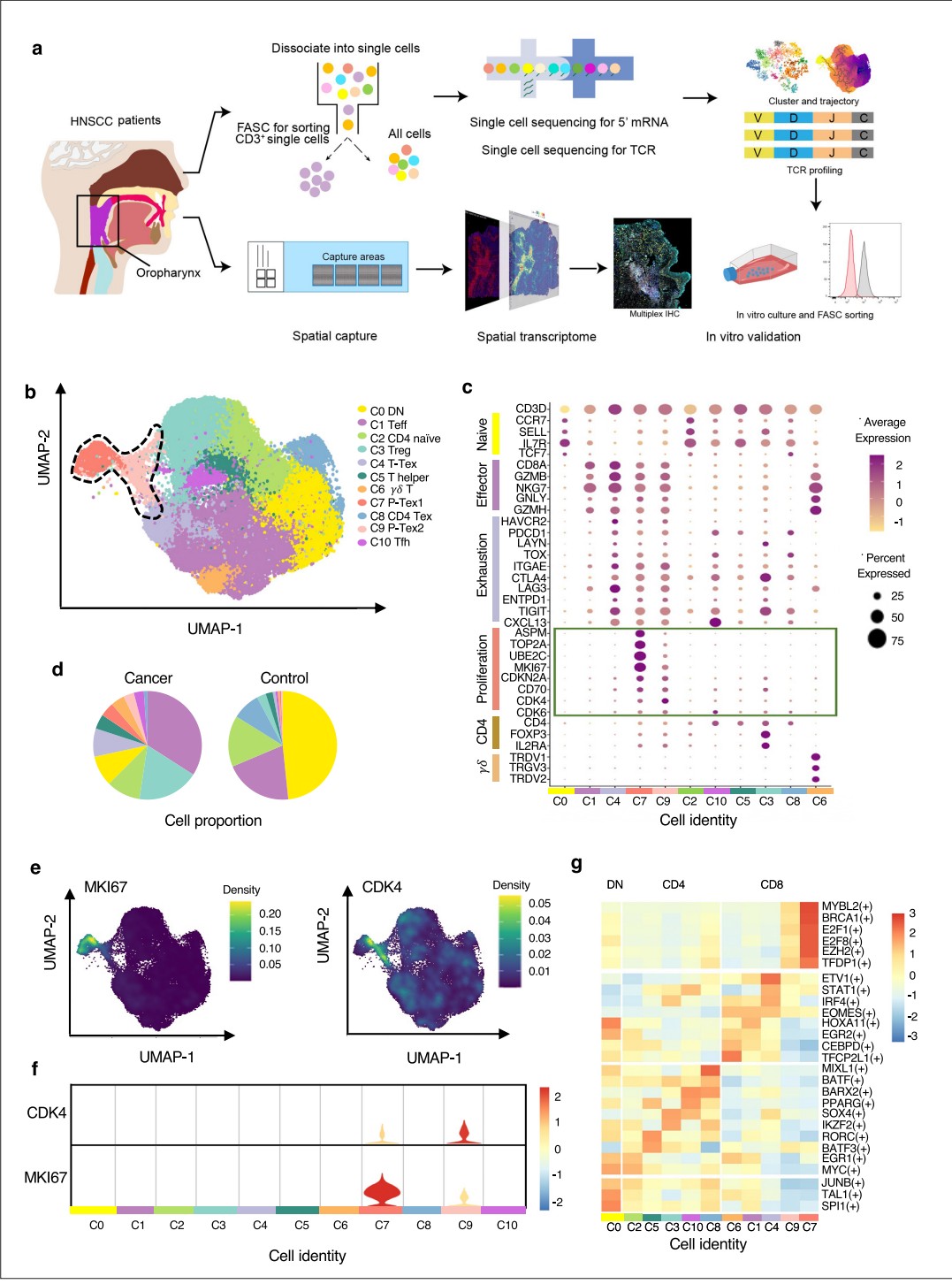

**Figure 1.** The P-Tex cell clusters identified in the T cell landscapes of head and neck squamous cell carcinoma (HNSCC) patients by single-cell RNA sequencing (scRNA-seq). (**a**) The flow chart of this study. (**b**) Uniform manifold approximation and projection (UMAP) plot of all single T cells from 14 samples via 10× Genomics. Eleven T cell clusters with different functions are identified. (**c**) Dotplot of selected T cell function-associated genes across different T cell clusters, showing both gene expression level (the color gradient) and the percentage of cells (the size of circle) in a given cluster. (**d**) Pie charts of cell-type fractions identified in cancer and normal adjacent tissues, colored by cell types. (**e**) The kernel density estimation plot showing the distribution of MKI67 and CDK4 genes of T cells. (**f**) The gene expression levels of CDK4 and MKI67 shown by violin plots. (**g**) Heatmap of the transcriptional regulators of top expressed genes in each T cell clusters.

*Figure 1 continued on next page*

*Figure 1 continued*

The online version of this article includes the following figure supplement(s) for figure 1:

**Figure supplement 1.** The extended summary of functional properties of T cell clusters in *Figure 1*.

T-Tex and Teff clusters (p<0.001, *Figure 2c*). To further investigate the cell cycle phase of each T cell cluster, we calculated cell cycle scores and visualized them on uniform manifold approximation and projection (UMAP) plots (*Figure 2d*, *Supplementary file 5*). P-Tex1 was mainly in G2/M phase, indicating cells were under a proliferative burst, which was consistent with the high expression of proliferation marker gene *MKI67* (*Bertoli et al., 2013*), whereas P-Tex2 was mainly in S phase, an essential phase for DNA replication before undergoing mitosis, which was consistent with the high expression of *CDK4* (initiating the G1 to S phase transition) (*Figure 2e–f*; *Bertoli et al., 2013*; *Romero-Pozuelo et al., 2020*; *Rubin et al., 2020*). Taken together, P-Tex1 and P-Tex2 might represent proliferation cells in two distinct cell cycle phases.

It is also noteworthy that HPV-positive HNSCC patients in TCGA (The Cancer Genome Atlas) cohort with higher P-Tex score (5-year OS: 55.8% vs. 22.6%, p=0.02), proliferation score (*MKI67*-related genes, 5-year OS: 49.1% vs. 15.8%, p<0.001), exhaustion score (*PDCD1*-related genes, 5-year OS: 56.2% vs. 23.1%, p=0.05), or cytotoxic score (*GZMB*-related genes, 5-year OS: 55.4% vs. 21.9%, p=0.02) had better survival outcomes, whereas similar trends were not observed in HPV-negative HNSCC patients (*Figure 2g–h*, *Supplementary file 5*). It is probably related to the difference in TME between HPV$^+$ vs. HPV$^-$ HNSCC.

Taken together, the P-Tex cluster displayed high expression levels of proliferation- and cycle-related genes. More importantly, HPV-positive HNSCC patients with higher P-Tex score, proliferation score, exhaustion score, or cytotoxic score had better survival outcomes, while this trend was not observed in HPV-negative HNSCC patients.

## Paired scRNA-seq and TCR-seq revealed the developmental trajectory of P-Tex

Given the clonal accumulation of CD8$^+$ T cells was a result of local T cell proliferation and activation in the tumor environment (*Tanchot et al., 2013*), we further conducted clonality analysis of CD8$^+$ T cells based on TCR-seq data. After quality control, we obtained TCRs with both alpha and beta chains for 33,897 T cells, including 20,607 unique TCRs, 2798 double TCRs, and 10,492 clonally expanded TCRs, with clonal sizes ranging from 2 to 162.

To further confirm whether the T cell clonality was associated with TME and HPV status, we systematically compared the clonality by cell clusters, tissue origin, and HPV status, respectively. The CD8$^+$ T cell clusters harbored more clonally expanded cells than CD4$^+$ T cell clusters and DN cell clusters in general, among which Tex harbored the highest proportions of clonal cells, followed by the two P-Tex clusters, which were more abundant than the Teff cells (*Figure 3a–b*, *Figure 3—figure supplement 1a–b*). Our results showed that hyperexpanded TCR clonotypes were more enriched in tumors than adjacent normal tissues (*Figure 3c–d*, *Figure 3—figure supplement 1c–d*). However, the proportions of hyperexpanded TCR clonotypes of Teff, P-Tex1, and T-Tex showed no significant difference between HPV-positive samples and HPV-negative samples (*Figure 3e–f*, *Figure 3—figure supplement 1e–f*). Correspondingly, a higher diversity of TCRs was observed in adjacent normal tissues and HPV-negative samples (*Figure 3g*), indicating the absence of a strong antigen-specific immune response, which further confirmed the crucial roles of virus and tumor play in local T cell proliferation and activation (*Hwang et al., 2020*; *Poropatich et al., 2017*).

We further examined the TCR clonotype occupation among each cluster and revealed that most of the shared TCRs were observed among the T-Tex, P-Tex, and Teff clusters (*Figure 3h–i*, *Supplementary files 6-7*). The Teff cluster had higher proportion of TCRs shared with the γδ T (overlap coefficient, oc = 0.49), Tex (oc = 0.43), P-Tex2 (oc = 0.35), and P-Tex1 (oc = 0.33) clusters, respectively, indicating they had common ancestry of origin. Besides, *Figure 3—figure supplement 1g–j* and *Figure 3—figure supplement 2a–c* show the distribution of the top shared clonotypes across CD8$^+$ T cell clusters, individuals, and HPV status. There were almost no shared TCRs across individuals, indicating the highly heterogeneous characteristics of T cells among individuals.

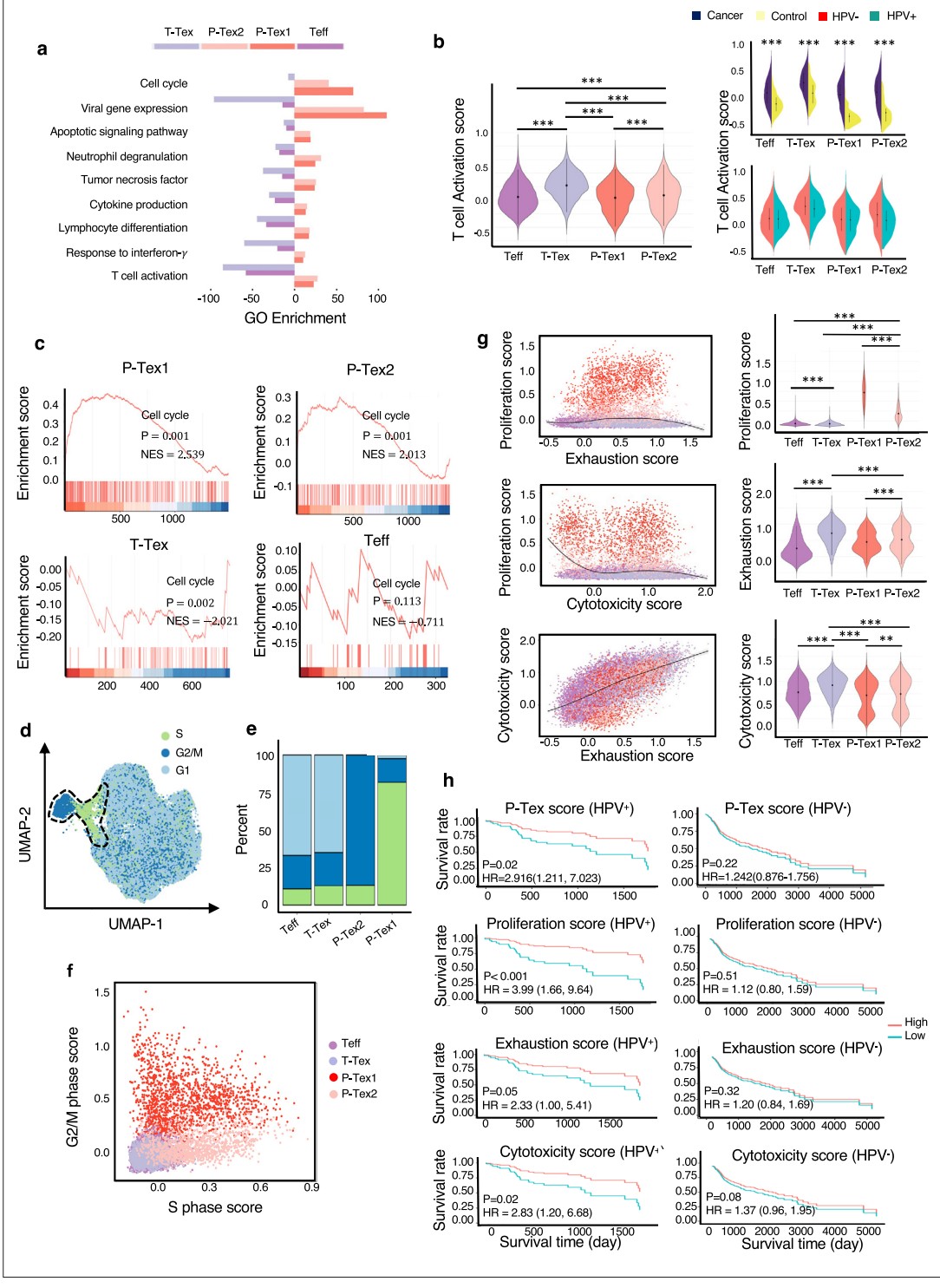

**Figure 2.** The comparison of functional characteristics between P-Texs and other CD8+ T cell clusters. (**a**) Gene ontology (GO) analysis of differentially expressed genes in each CD8+ T cell clusters, colored by cell types. (**b**) The T cell activation score of each CD8+ T cell clusters. T cell activation score defined as the averaged expression of genes in the activation signature in ***Supplementary file 3***. Statistics were assessed by Dunn's tests. (**c**) The gene set enrichment analysis (GSEA) diagrams show the enrichment profiles of cell cycle pathway in each CD8+ T cell clusters. (**d–f**) The distribution and scores of cell cycle phases of each CD8+ T cells. (**g**) The proliferation, exhaustion, and cytotoxic scores of each CD8+ T cell clusters. Proliferation score: averaged expression of MKI67-related genes; exhaustion score: averaged expression of PDCD1-related genes; cytotoxic score: averaged

*Figure 2 continued on next page*

*Figure 2 continued*

expression of GZMB-related genes. (**h**) The Kaplan–Meier curves show the overall survival rate of HPV$^+$/HPV$^-$ head and neck squamous cell carcinoma (HNSCC) patients with different proliferation, exhaustion, and cytotoxic scores in The Cancer Genome Atlas (TCGA) cohort, adjusted for age and gender. \*\*\*: p<0.001, \*\*: p<0.01, \*: p<0.05.

To further investigate their lineage relationships, we performed pseudotime analysis for CD3$^+$ T cells on the basis of transcriptional similarities (*Figure 3j–l*, *Figure 3—figure supplement 2d*). The starting point of pseudotime was the DN cluster, with CD4$^+$ T cell and CD8$^+$ T cell clusters differentiating toward two different directions, suggesting of their distinct developmental paths. Notably, two P-tex clusters primarily aggregated at the end of the pseudotime backbone of CD8$^+$ T cells, and presented to be a specific branch originating from T-Tex, demonstrating its specific activation state and characteristic, which was distinct from other T-Tex cells. Besides, P-Tex2 was located ahead of P-Tex1 on the pseudotime, which was consistent with the results in *Figure 2d–e*, where P-Tex2 cells mainly entered the S phase of the cell cycle (early phase), while P-Tex1 mainly entered the G2/M phase (later phase).

Taken together, given that two P-Tex clusters were located at the end of the developmental trajectory of the Teff and T-Tex cells and that P-Tex clusters partially shared TCRs with the T-Tex cluster, we speculated that Teff cells transformed from an activated to exhausted state (T-Tex cells), and some of the T-Tex cells could further gradually transform into a unique P-Tex subpopulation with a highly specific proliferation state.

## The survival, proliferation capacity, and cytotoxicity of P-Texs in vitro

To further verify the aforementioned function of P-Tex cells, we sorted P-Tex cells (CD3$^+$CD8$^+$U-BE2C$^+$PD1$^+$) and T-Tex cells (CD3$^+$CD8$^+$UBE2C$^-$KLRB1$^+$PD1$^+$) via flow cytometry to compare their functions in vitro (*Figure 4a–g*). Unexpectedly, our flow cytometry results showed that the proliferation rate of P-Tex cells cultured with IL-2 for 15 and 20 days in vitro was much slower than that of T-Tex cells (*Figure 4a*), whereas the cell viability of P-Tex cells was much higher than that of T-Tex cells (*Figure 4b–c*), indicating that instead of showing high proliferation capacity when stimulated in vitro, the P-Tex cells were mainly characterized by prolonged cell survival.

Moreover, P-Tex cells expressed higher levels of proliferation-related marker UBE2C, as measured by flow cytometry, and the variations of UBE2C in P-Tex and T-Tex cells between 9 and 16 days were relatively stable (*Figure 4d–g*). Besides, compared with T-Tex cells, P-Tex cells expressed higher exhaustion-related markers (PD1), and the expression of PD1 in P-Tex and T-Tex cells gradually increased from day 9 to day 16. Meanwhile, a larger proportion of T-Tex cells produced more cytotoxic-related markers (KLRB1) than P-Tex cells after stimulated with CD3/CD28 microbeads and IL-2 for 9 and 16 days in vitro, whereas the expression of KLRB1 in T-Tex gradually decreased since day 9, and the expression of KLRB1 in P-Tex was relatively stable. Meanwhile, the results of our in vitro experiments were consistent with the aforementioned findings showing that the T-Tex cells were more activated than those of P-Tex cells (*Figure 4h*).

Taken together, P-Tex cells represent a unique subcluster of the exhausted CD8$^+$ T cells, which was characterized by prolonged cell survival in vitro and could provide modest but persistent anti-tumor effects.

## The effect of CDK4 inhibitor on P-Tex cells might be a reason for its ineffectiveness in HPV-positive HNSCC patients

To better understand the anti-tumor role of P-Tex within the TME, we additionally conducted 5' droplet-based scRNA-seq profiles (10× Genomics) for primary tumors with paired adjacent normal tissues from two HNSCC patients. All biopsies were histologically examined by two independent pathologists. After quality control, a total of 13,515 cells from tumors (9040 cells) and adjacent normal tissues (4476 cells) were obtained. Given the fact that higher heterogeneity of cellular compositions exists in the TME than pure T cells, we recategorized all cells into 20-cell clusters according to previously reported markers (*Figure 5a*, *Figure 5—figure supplement 1a*, *Supplementary file 8*). We consistently identified that P-Tex cells highly expressed proliferation- and cell cycle-related genes and functions as the cancer epithelial cluster (*Figure 5b*, *Figure 5—figure supplement 1b–d*).

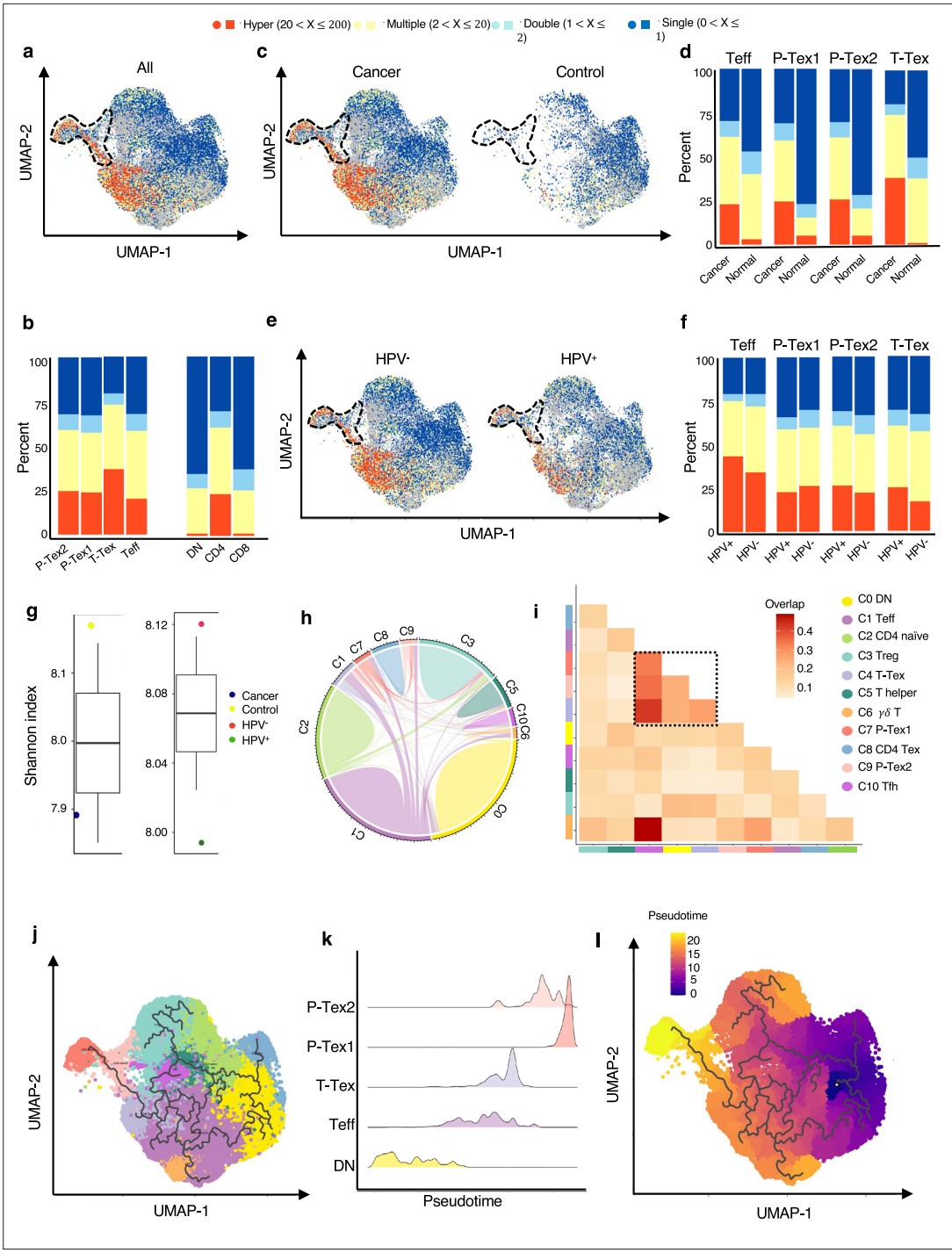

**Figure 3.** The developmental trajectory and lineage relationships among T cell clusters. (**a–f**) Single-cell TCR profiling of head and neck squamous cell carcinoma (HNSCC) in all samples (**a–b**), cancer tissues vs. normal tissues (**c–d**), HPV⁺ vs. HPV⁻ (**e–f**). Bar plots show the fractions of each clonotype frequencies. The clonotype frequencies are defined as unique (n=1), double (n=2), multiple clones (2 < n ≤ 20), and hyper clones (20 < n ≤ 200) according to the numbers of clonotypes. (**g**) The TCR diversity of cancer tissues vs. normal tissues and HPV⁺ vs. HPV⁻ samples, calculated using Shannon metric. (**h–i**) Cell state transition of T cell clusters inferred by shared TCRs. The chord diagram (**h**) showing the fraction of shared clonotypes among each cell clusters. Lines connecting different clusters are based on the degree of TCR sharing, with the width of lines representing the number of shared TCRs. The clonal overlap diagram (**i**) measures the clonal similarity among each cluster. Color gradient in the grid refers to

*Figure 3 continued on next page*

*Figure 3 continued*

the overlap coefficient. The higher the index score, the higher the clonal diversity. (**j–l**) Potential developmental trajectory of T cells inferred by Monocle3 based on gene expressions.

The online version of this article includes the following figure supplement(s) for figure 3:

**Figure supplement 1.** Extended summary of TCR properties of *Figure 3*.

**Figure supplement 2.** Extended summary of TCR clonotypes and potential differentiation direction of *Figure 3*.

P-Tex clusters were predominantly tumor-derived (*Figure 5—figure supplement 2a–b*) cells. We further investigated the expression and distribution of proliferation-related (*CDK4*, *MKI67*) and cancer-related epithelial (*KRT15*, *CD24*) cell marker genes among each cluster (*Figure 5c*). Notably, *CDK4* was highly expressed in P-Tex2 cells, cancer epithelial cells, and fibroblasts, while *MKI67* was highly expressed in P-Tex1 clusters. CDK4 is a well-known cancer treatment target, and CDK4 inhibitors (e.g., abemaciclib and palbociclib) have demonstrated cytostatic activity in HPV-negative HNSCC, whereas their effects on HPV-positive HNSCC are not obvious (*Adkins et al., 2019*; *Oppelt et al., 2021*; *Robinson et al., 2019*). And it was interesting that HPV-positive HNSCC patients with higher *CDK4* expression levels showed better survival than patients with lower *CDK4* expression, whereas HPV-negative HNSCC patients with higher *CDK4* expression levels showed worse prognosis (*Figure 5d*). Besides, compared with HPV-negative patients, the proportion of P-Tex was higher in the TME of HPV-positive HNSCC patients (TCGA cohort, *Figure 5e*, *Figure 5—figure supplement 2c*, *Supplementary file 9*). These findings raised the question of whether P-Tex cells that were beneficial to the prognosis of HPV-positive HNSCC patients would be simultaneously suppressed by CDK4 inhibitors.

To answer this question, we compared the cell viability of P-Tex cells and cancer epithelial cells by culturing with abemaciclib in vitro, respectively. As expected, abemaciclib inhibited the cell viability of both cancer cells (FD-LSC-1 cells) and P-Tex cells (*Figure 5f*). Therefore, we speculated that the inhibition of CDK4 inhibitor on the cell viability of P-Tex cells (which were beneficial to the survival prognosis of HPV-positive HNSCC patients) might be a potential reason why CDK4 inhibitors were ineffective in treating HPV-positive HNSCC patients.

## The cell–cell interactions between T cells and APCs in the HNSCC TME

To determine the underlying mechanism by which P-Tex cells fulfill their proliferation-related and anti-tumor function, we systematically explored the crosstalk between T cells and other cells in the HNSCC TME. The results showed that the interactions between P-Tex and Tex clusters and APCs (especially DCs) in the HNSCC TME were mainly enriched in T cell activation and proliferation signaling pathways, such as CD70-, CD80-, ICOS-, and PD-L1-related signaling pathways (*Figure 6a–b*, *Figure 6—figure supplement 1* and *Supplementary file 10*). Given the fact that the co-localization of APCs and T cells is the precondition for fulfilling their function related to antigen presentation and T cell activation, we further conducted spatial transcriptome (ST) analysis for representative fresh HNSCC tumor (P_08) to verify their spatial distribution characteristics. We identified 17 spatial cluster areas, among which cluster 16 were defined as APC area (*Figure 6c*). The P-Tex and Tex cells were characterized by the co-localization in the APC aggregation area with significantly higher P-Tex scores and Tex scores than other non-APC areas (p<0.001), and the correlations among the three scores in the APC area were higher than those in other non-APC areas. As expected, the activation score of T cells were higher in APC area (*Figure 6d–e*, *Figure 6—figure supplement 2*, *Supplementary file 11*). Besides, the aforementioned ligand–receptor interactions of T cell activation and proliferation signaling pathways (CD70-CD27, CD80-ICOS, CD86-CTLA4, CD274-PDCD1) were also detected in the APC areas (*Figure 6f*). We also observed enriched signaling pathways in APC areas involving the cell cycle, neutrophil activation, and RNA splicing (*Figure 6g*), supporting that these antigen-presenting cells play a role in modulating the immune response within the TME by promoting T cell activation (*Mehrfeld et al., 2018*; *Wosen et al., 2018*).

To further confirm the ST results (transcriptomic level) at the proteomic level, we performed multiplex immunofluorescence (mIF) of both canonical APC markers (MHC-II[+]) and selected markers for P-Tex cells (CD8[+]PD1[+]CDK4[+]/MKI67[+]), bulk Tex cells (CD8[+]PD1[+]CDK4[-]MKI67[-]) and bulk CD8[+] T cells (CD8[+]) on formalin-fixed paraffin-embedded (FFPE) tissue originating from HNSCC patients (*Figure 7a–b*). Next, we explored average distances of APCs to P-Tex cells, bulk Tex cells, and bulk

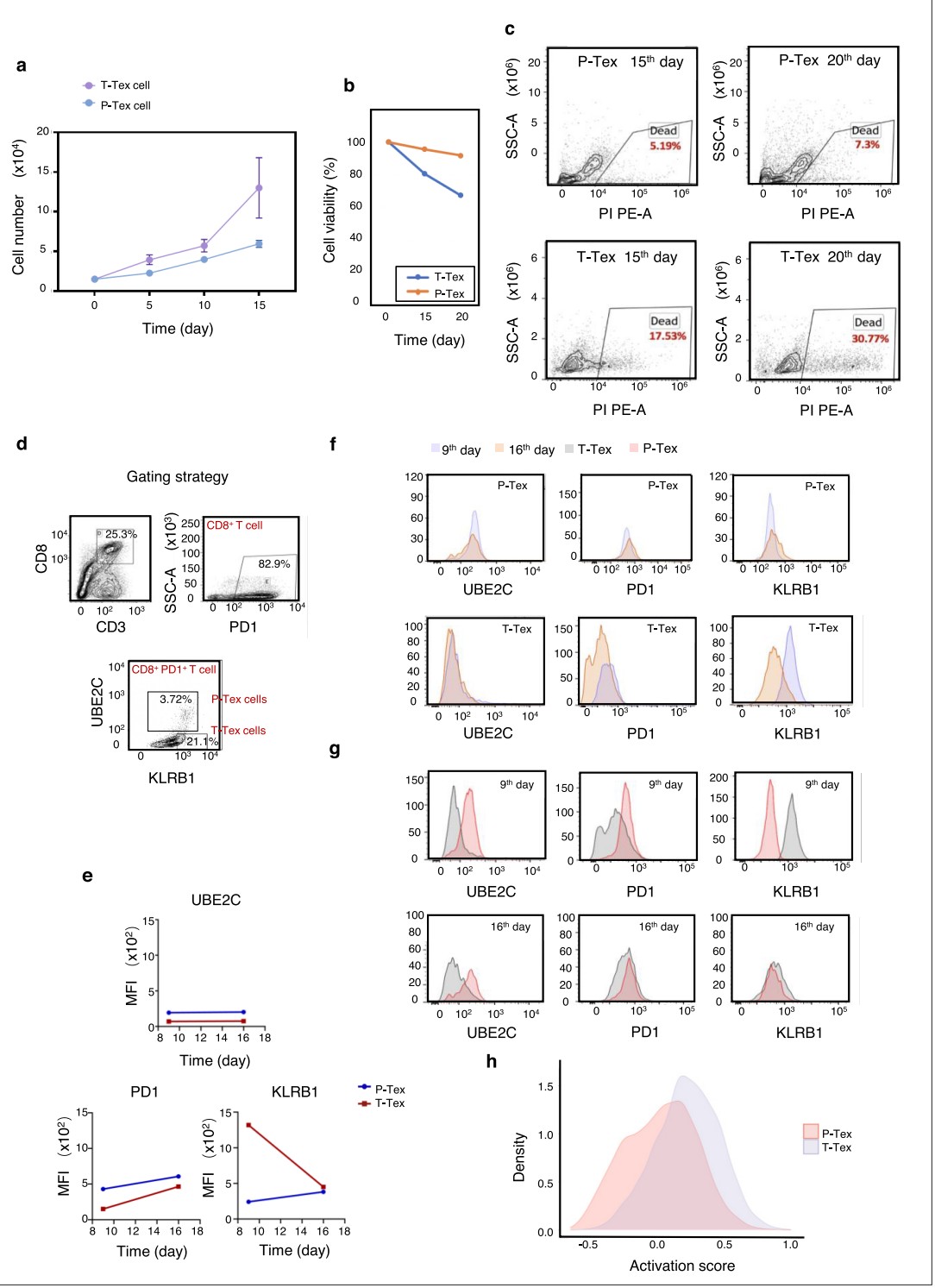

**Figure 4.** The survivaland proliferation capacity of P-Texs in vitro. Comparing the proliferation (**a**) and survival (**b–c**) capacity of P-Tex and T-Tex cells cultured with IL-2 for 15 and 20days in vitro measured by CCK8 experiment; proliferation experiment, shown by the mean ± SEM. (**d–g**) Representative flow cytometry assay of UBE2C, PD1, and KLRB1 of P-Tex cells after 9 and 15 days of stimulation with anti-CD3/CD28 microbeads in vitro. (**d**) The gating strategies of PD1, KLRB1, and UBE2C. (**e–g**) Mean fluorescence intensity (**e**) and cell count (**f–g**) of UBE2C, PD1, and KLRB1 in P-Tex and T-Tex cells at different days detected by flow cytometry. (**h**) Histogram of activation states of P-Tex and Tex cells by single-cell RNA-seq data.

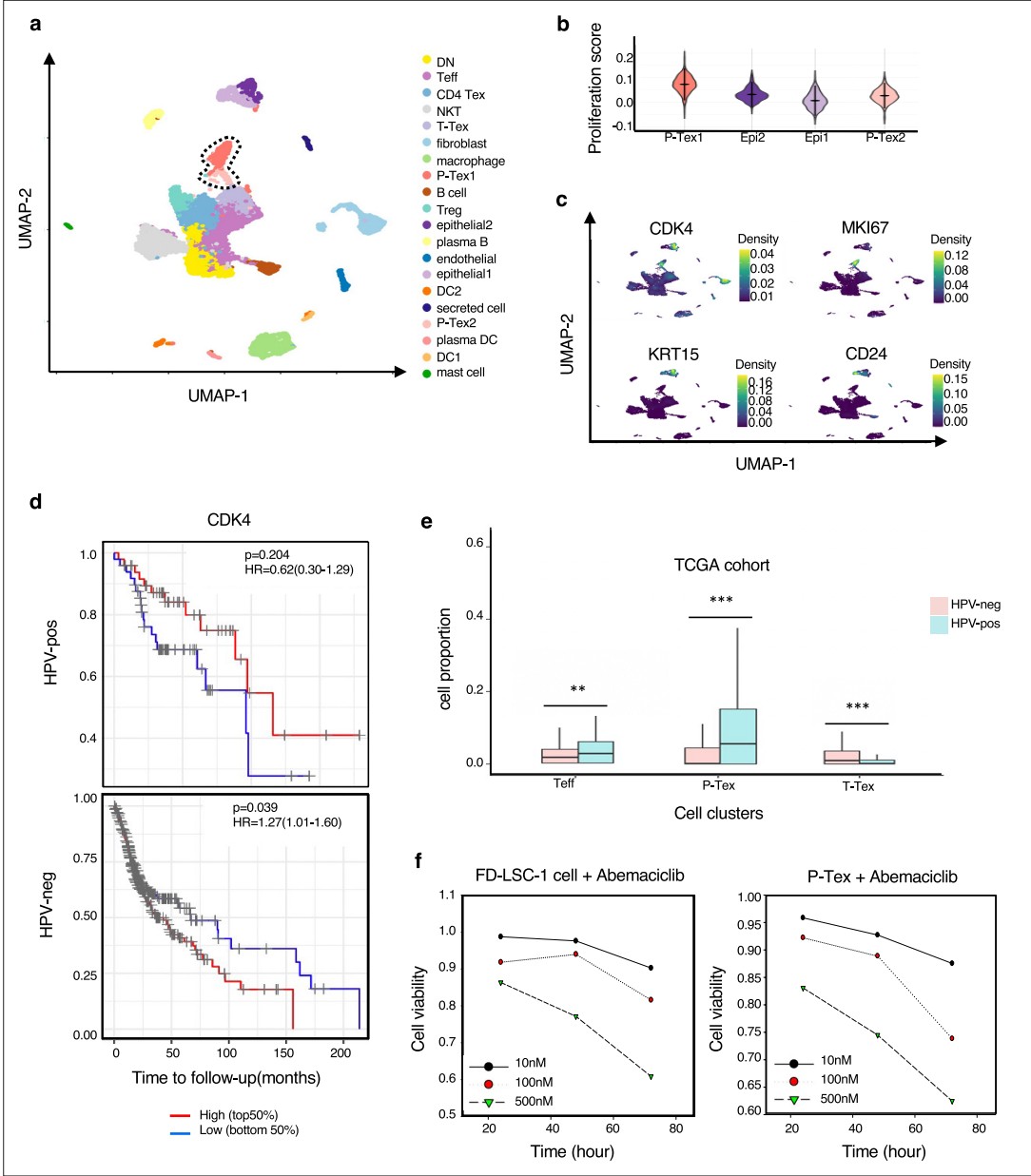

**Figure 5.** The expression of CDK4 gene in P-Tex2 cluster is associated with the treatment outcomes of HPV[+] head and neck squamous cell carcinoma (HNSCC) patients. (**a**) Single-cell transcriptomic profiling of HNSCC tumor microenvironment (TME). Twenty cell clusters are identified, colored by cell types. (**b**) The proliferation status of P-Tex and epithelial cells in violin plot. (**c**) The kernel density estimate distribution of proliferation markers (CDK4 and MKI67) and epithelial cancer cell markers (KRT15 and CD24) in uniform manifold approximation and projection (UMAP) plots. (**d**) The overall survival rate of HPV[+]/HPV[-] HNSCC patients in The Cancer Genome Atlas (TCGA) cohort related to the expression levels of CDK4 gene, adjusted for age and gender. (**e**) The proportion of P-Texs, T-Tex, and TEFF clusters in HPV[+] and HPV[-] samples in TCGA cohort by using the deconvolution algorithm; statistics were assessed by Chi-square tests. Marker genes that were used to define cell clusters in (**a**) are deconvolved into the TCGA data to obtain the proportion of P-Texs, T-Tex, and Teff clusters in the TCGA cohort. (**f**) The cell viability of P-Tex and cancer epithelial cells assessed by CCK8 experiment after Abemaciclib treated in vitro. ***: p<0.001, **: p<0.01, *: p<0.05.

The online version of this article includes the following figure supplement(s) for figure 5:

**Figure supplement 1.** The extended summary of functional properties of cell clusters in *Figure 5*.

**Figure supplement 2.** The extended summary of function properties of cell clusters in *Figure 5*.

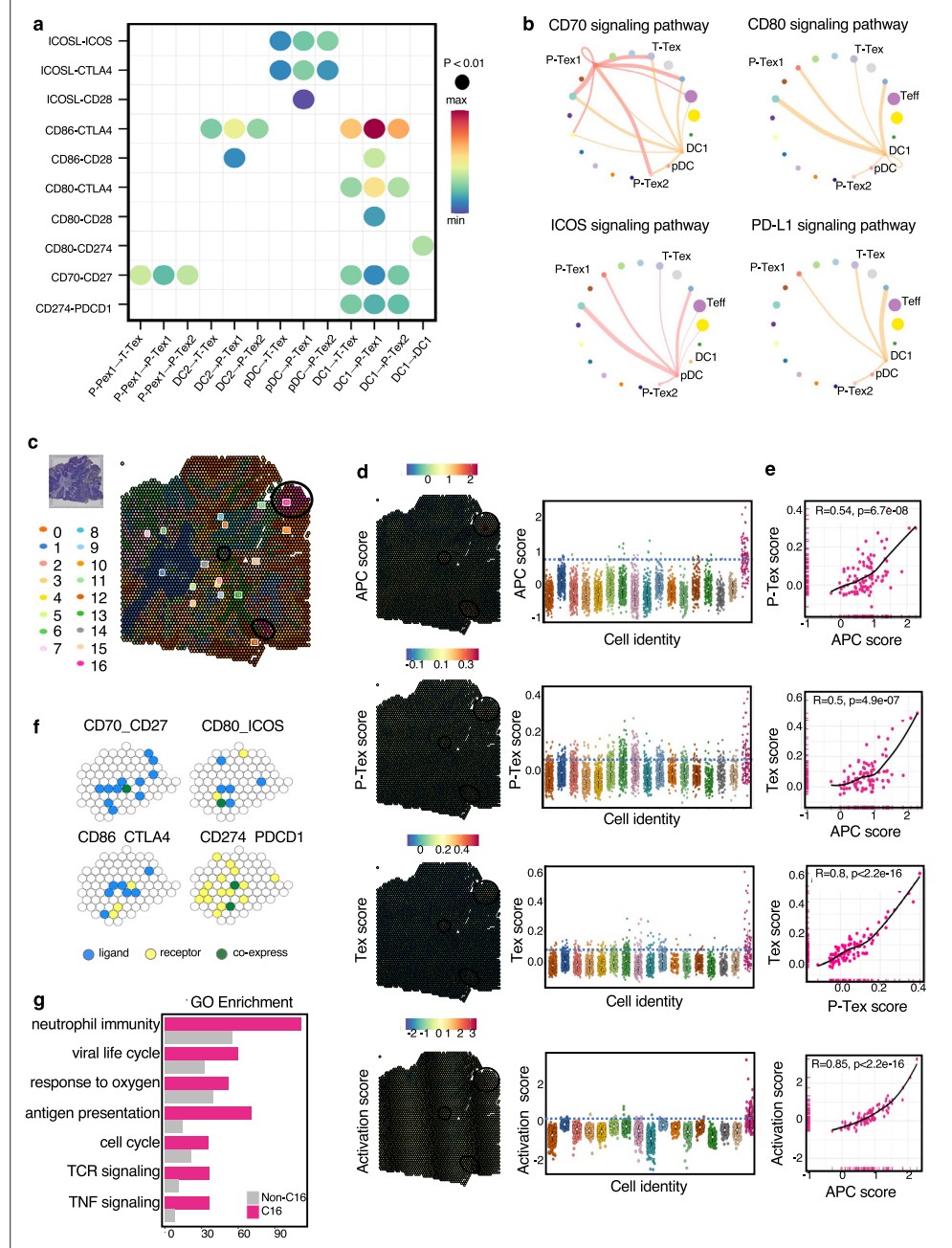

**Figure 6.** The cell–cell interactions between T cells and APC cells are enriched in the proliferation and cell activation pathways in head and neck squamous cell carcinoma (HNSCC) tumor microenvironment (TME). (**a**) The communication probabilities mediated by selected ligand–receptor pairs among different cell types. The color gradient shows the level of interaction. (**b**) Network circle graphs visualize the inferred communication network of signaling pathways among different cell clusters derived by ligand–receptor interactions. The color of lines are consistent with the ligands. The width of lines are proportional to the interaction strength, and the circle sizes are proportional to the number of cells in each clusters. (**c**) The spatial transcriptomic landscape of representative HNSCC samples. (**d–e**) P-Tex and Tex features were co-expressed in APC area (cluster 16). The circles in SpatialDimPlot (**d**, left) represent APC, P-Tex, and Tex scores enriched in the APC area (cluster 16). The Texs and P-Tex scores were higher in the APC aggregation area (**d**, right). The correlation of P-Tex, Tex, activation scores, and APC scores in the spatial transcriptome (**e**). (**f**) Spatial feature plots of selected ligands–receptor interactions enriched in APC area. Spatial feature plots showing the expression pattern of single ligand genes (CD70, CD80, CD86, CD274, yellow spots), single receptor genes (CD27, ICOS, CTLA4, PDCD1, blue spots), and co-expression

*Figure 6 continued on next page*

*Figure 6 continued*

pattern (green spots) in APC area. (**g**) Gene ontology (GO) analysis identified the enriched gene functions in APC area of spatial transcriptome.

The online version of this article includes the following figure supplement(s) for figure 6:

**Figure supplement 1.** The supplementary cell–cell interactions of head and neck squamous cell carcinoma (HNSCC) tumor microenvironment (TME) for *Figure 6b*.

**Figure supplement 2.** The correlation of Tex, P-Tex, and activation scores with APC scores for each cluster in spatial transcriptomics.

CD8[+] T cells (*Figure 7c*), respectively. As expected, 90% of the P-Tex cells, bulk Tex, and bulk CD8[+] cells enriched within a distance of 25 μm from APCs, forming an intra-tumoral niche (*Jansen et al., 2019*).

Taken together, P-Tex cells were enriched in the APC aggregation areas, and the signal pathways related to T cell activation and proliferation were activated in these areas, indicating that P-Tex might act as a specific T cell pool that provides modest but persistent effects through its interactions with APCs.

## Discussion

Our current study provided a comprehensive multi-omics characterization of over 49,000 tumor-infiltrating CD3[+] T cells in HNSCC patients. A special novel P-Tex cluster that expressed high levels of proliferation- and cell cycle-related genes as well as cytotoxic and checkpoint molecules was identified. Our results showed that HPV-positive HNSCC patients who had higher proportions of P-Tex cells had a better survival prognosis. Unexpectedly, we also found that P-Tex cells expressed CDK4 genes as high as cancer cells, which could be simultaneously inhibited by the CDK4 inhibitors. Therefore, we speculated that it might be a potential reason for the ineffectiveness of CDK4 inhibitors in treating HPV-positive HNSCC. Furthermore, P-Tex cells were fund to be aggregated in the APC areas where their T cell activation and proliferation signaling pathways were activated. Together, our findings reveal a promising role for P-Tex cells in the prognosis of HPV-positive HNSCC patients by providing modest but persistent anti-tumor effects.

There is accumulating evidences showing that heterogeneity is a hallmark of T cell exhaustion, and a typical three-stage differentiation trajectory (progenitor–transitional–terminal) has been established to depict the corresponding spatiotemporally sequential alterations of gene signatures, functional characteristics, as well as epigenetic modifications (*Beltra et al., 2020*; *Im and Ha, 2020*). Despite several previous studies have identified similar proliferation Tex clusters in chronic LCMV-infected mouse models using scRNA-seq, little attention was paid to their potential roles in anti-tumor immunity (*Miller et al., 2019*; *Wagle et al., 2021*; *Yao et al., 2019*). In this study, we systematically investigated the functional characteristics and developmental trajectory of P-Tex cells by comparing with other CD8[+] T cell clusters. Our results suggested that P-Tex was an independent branch of Tex cells and might act as a T cell pool, providing modest but persistent anti-tumor immunity through its prolonged cell survival and highly specialized cytotoxic capacity. However, this beneficial on long-term survival outcomes was only observed in HPV-positive HNSCC who had higher proportion of P-Tex.

CDK4/6 inhibitors (e.g., palbociclib, ribociclib, and abemaciclib) are promising drugs for various cancers (*Du et al., 2020*), working by specifically inhibiting CDK4/6 proteins, blocking the transition from the G1 to the S phase of the cell cycle and preventing cancer cell progression (*Álvarez-Fernández and Malumbres, 2020*). Notably, *CDK4,* which was highly expressed in cancer cells, was also found to be highly expressed in the P-Tex cells. Our in vitro results showed that CDK4 inhibitors could simultaneously inhibit the cell viability of both cancer cells and P-Tex cells. Due to the fact that P-Tex cells were benefit to the prognosis of HPV-positive HNSCC patients, we speculated that the inhibition of CDK4 inhibitors on P-Tex cells might be one of the reasons why promising treatment outcomes of CDK4 inhibitors were not observed in HPV-positive HNSCC patients (*Adkins et al., 2019*; *Oppelt et al., 2021*; *van Caloen and Machiels, 2019*).

Overall, a novel promising P-Tex cluster, which was mainly identified in APC areas of TME, was beneficial to the survival prognosis of HPV-positive HNSCC,. Besides, the inhibitory effect of CDK4 inhibitors on P-Tex cells helps clarify its ineffectiveness in HPV-positive HNSCC patients.

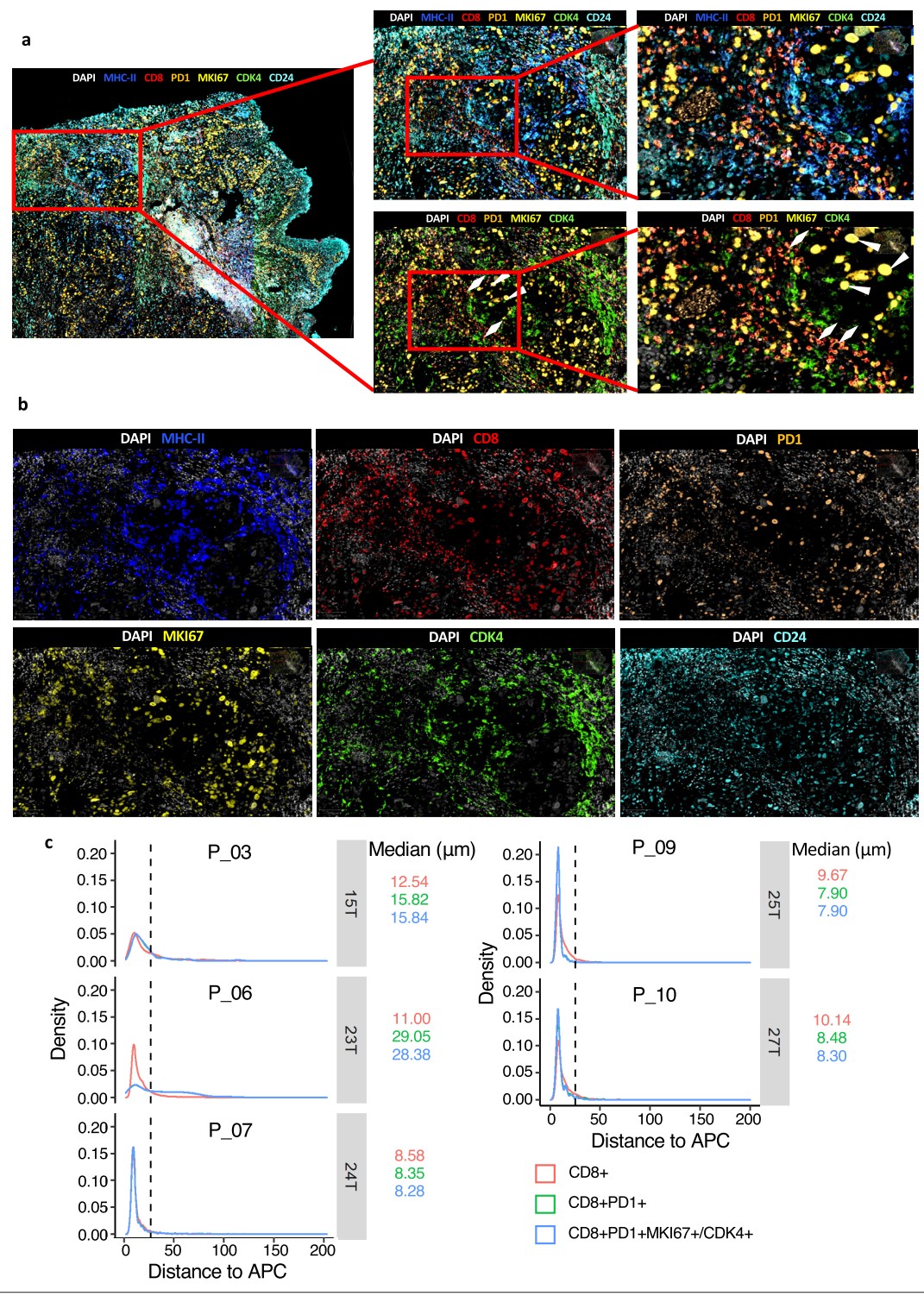

**Figure 7.** The spatial characteristics of APC, pro-Tex cells, and Tex cells in the head and neck squamous cell carcinoma (HNSCC) tumor microenvironment (TME). (**a–b**) Representative example of HNSCC tumor stained by multiplex IHC, with white triangles and rhombus showing the Texs and P-Tex aggregates in the APC area, respectively. (**c**) Measured distances to APC (MHCII⁺) cells from CD8⁺ T cells, CD8⁺ PD1⁺T cells (Tex cells) or CD8⁺ PD1⁺MKI67⁺/CDK4⁺ T cells (P-Tex cells) in five representative samples. The dashed lines represent the cutoff distance of 25 µm, which indicate that 90% of CD8⁺ T, Tex, or P-Tex cells are enriched within a distance of 25 µm from APCs.

# Materials and methods

**Key resources table**

| Reagent type (species) or resource | Designation | Source or reference | Identifiers | Additional information |
|---|---|---|---|---|
| Sequence-based reagent | Chromium Single Cell 5' Gel Bead and Library Construction Kit | 10× Genomics | PN-1000006; PN-1000020 | |
| Sequence-based reagent | Single Cell V(D)J Enrichment Kit Human T cell | 10× Genomics | PN-1000005 | |
| Sequence-based reagent | Visum Spatial Library Construction kit | 10× Genomics | PN-1000184 | |
| Antibody | FITC Mouse Anti-Human CD3 (mouse monoclonal) | BD Pharmingen | Cat#555332 | FACS (0.8 µL per test) |
| Antibody | PE anti-human CD8a Antibody (mouse monoclonal) | Cell Signaling Technology | Cat#300908, Clone: HIT8a | FACS (5 µL per test) |
| Antibody | Anti-human CD279 (PD-1) (mouse monoclonal) | Biolegend | Cat#329920 | FACS (5 µL per test) |
| Antibody | UBE2C (B-8) PE (mouse monoclonal) | Santacruz | Cat#Sc271050 | FACS (1:100) |
| Antibody | PE/Cyanine7 anti-human CD161 (mouse monoclonal) | Biolegend | Cat#339917 | FACS (5 µL per test) |
| Antibody | Anti-CD8α (mouse monoclonal) | Cell Signaling Technology | Cat#70306S | IHC (1:200) |
| Antibody | Anti-PD-1 (Rabbit monoclonal) | Cell Signaling Technology | Cat#84,651T | IHC (1:200) |
| Antibody | Anti-Ki67 (Rabbit monoclonal) | Abcam | Cat#ab16667, Clone: SP6 | IHC (1:50) |
| Antibody | Anti-CDK4 (Rabbit monoclonal) | Cell Signaling Technology | Cat#12790 | IHC (1:1000) |
| Antibody | Anti-CD24 (Rabbit Polyclonal) | Proteintech | Cat#10600-1-AP | IHC (1:200) |
| Antibody | Anti-HLA-DR (Rabbit monoclonal) | Abcam | Cat#ab92511 | IHC (1:200) |
| Antibody | Anti-EpCAM (Rabbit monoclonal) | Abcam | Cat#ab223582, Clone: EPR20532-225 | IHC (1:500) |
| Commercial assay or kit | Cell Counting Kit-8 | MedChemExpress | Cat#HY-K0301 | |
| Commercial assay or kit | Manual Opal 7-Color IHC Kit | Akoyabio | Cat#NEL811001KT | |
| Cell line (*Homo sapiens*) | Laryngeal carcinoma | This paper | FD-LSC-1 | Cell line maintained in State Key Laboratory of Biotherapy, West China Medical School, Sichuan University |
| Chemical compound, drug | Abemaciclib (LY2835219) | Topscience | Cat#T2381 | |
| Software, algorithm | Cell Ranger count | 10× Genomics | v3.1.0 | |
| Software, algorithm | Cell Ranger VDJ | 10× Genomics | v6.1.1 | |
| Software, algorithm | Space Ranger | 10× Genomics | v2.0 | |
| Software, algorithm | Seurat | R | RRID:SCR_007322 | |
| Software, algorithm | monocle3 | R | RRID:SCR_018685 | |
| Software, algorithm | CellChat | R | RRID:SCR_021946 | |

*Continued on next page*

*Continued*

| Reagent type (species) or resource | Designation | Source or reference | Identifiers | Additional information |
|---|---|---|---|---|
| Software, algorithm | pheatmap | R | RRID:SCR_016418 | |
| Software, algorithm | FlowJo | FlowJo | RRID:SCR_008520 | |

## Ethical statement

This study was conducted in accordance with the Declaration of Helsinki (as revised in 2013) and was approved by the Biomedical Research Ethics Committee of West China Hospital (2021-908), with the individual consent for each participant.

## Specimen collection and processing

Patient's information was summarized in *Supplementary file 1*. HNSCC tumor tissues and paired adjacent normal tissues were collected during the surgery. Then, the tissues were rinsed by 1× PBS, with surrounding necrotic areas being carefully removed and were cut into small pieces of 2–4 mm and reserved in the mixture of 1× DMEM medium (Gibco) and penicillin-streptomycin solution (Hyclone). The remaining tissues were fixed into FFPE blocks for HE staining and mIF.

## Preparation of single-cell suspensions

The tissue pieces were rapidly transferred into the gentleMACS C Tube containing Human Tumor Dissociation Kit (Miltenyi Biotec, #130-095-929) according to the manufacturer's recommendation. The dissociated cells were filtered through 40 µm cell strainers to remove clumps. Cell pellets were resuspended in binding buffer after centrifuged and sorted via human CD3 MicroBeads (Miltenyi Biotec, #130-050-101) according to the manufacturer's recommendation (Note: as for the experiment of scRNA-seq of overall cells, CD3 sorting was not needed). The overall cells and the sorted CD3$^+$ T cells were separately resuspended in HBSS (Gibco) plus 0.04% bovine serum albumin (Sigma-Aldrich) and tested for cell viability. Cell viability >80% was required for subsequent library constructions.

## Library construction and sequencing

ScRNA-seq was performed using Chromium Single Cell 5' Gel Bead and Library Construction Kit (10× Genomics, #PN-1000006, PN-1000020) and Single Cell V(D)J Enrichment Kit Human T cell (10× Genomics, #PN-1000005). Reverse transcription, cDNA recovery, cDNA amplification, and library construction were performed according to the manufacturer's protocol. The constructed libraries were sequenced on NovaSeq 6000 (Illumina) with paired-end sequencing and single indexing.

## Quality control and preprocessing of sequencing data

Cell Ranger count (v3.0) (*Zheng et al., 2017*) was used to process the raw FASTQ files, align the sequencing reads to Ensembl GRCh38 reference genome (http://cf.10xgenomics.com/supp/cell-exp/refdata-cellranger-GRCh38-3.0.0.tar.gz), and exclude background noise to generate a filtered UMI expression matrix for each cell. The package Seurat (v4.0.4) (*Hao et al., 2021*) were used to filter cells that were empty droplets or doublets and that have >5% mitochondrial counts. Next, we normalized the expression matrix via 'LogNormalize' and log-transform method. Then, we apply a linear transformation to prepare the expression matrix for next step dimensional reduction.

## Unsupervised clustering of cells and UMAP visualization

The high cell-to-cell variable features (top 2000) between cells were used as input to perform principal component analysis on the scaled matrix. Subsequently, we employed Harmony (v1.0, R package) (*Korsunsky et al., 2019*) to integrate multiple samples and the top 30 dimensions were selected for UMAP with the reduction of 'harmony'.

We performed Seurat to cluster cells using the Louvain algorithm. The previous reported marker genes (*Supplementary file 3*) were used for the cell cluster annotation with gene functional description and gene expression. Nebulosa (v1.3.0, R package) (*Alquicira-Hernandez and Powell, 2021*) was applied to perform gene kernel density estimation and visualize cell features on UMAP plot.

## Differential expression and analysis of signaling pathways

To characterize the function of defined clusters, we used Seurat to calculate differentially expressed genes (DEGs) among each cluster, identified marker genes as DEGs with adjusted p-value <0.05 and put marker genes into clusterProfiler (v4.0.2, R package) (*Wu et al., 2021*) to perform GO enrichment analysis (p<0.01) and GSEA and visualization.

## Transcription factor regulatory network analysis

To predict the gene regulatory network within cell clusters, we used previous selected top 2000 variable features-barcode matrix from scRNA-seq data as input and applied pySCENIC (v0.11.2) (*Aibar et al., 2017*) to infer the network activity in each cell cluster.

## Cell score calculation

We applied AddModuleScore function embedded in Seurat to calculate the specific cell scores in different clusters, which was defined as: the average gene expression of specific gene panel in each cluster, subtract the average gene expression of random control gene sets (*Puram et al., 2017*; *Supplementary file 3*). Functional module scores were based on the expression levels of top 30 genes which were highly correlated with GZMB (cytotoxicity score), PDCD1 (exhaustion score), or MKI67 (proliferation), respectively. TCR-dependent T cell activation score was calculated based on the activation gene signature (*Azizi et al., 2018*). Proliferation score was calculated based on the genes enriched in the GO molecular function term of 'cell cycle phase transition'. The specific cluster score (P-Tex, Tex, and APC score) was calculated based on marker genes of each cluster listed in *Supplementary file 3*.

We assigned cell cycle scores based on the expression of G2/M and S phase marker genes and predicted the classification of each cell in either G2/M, S, or G1 phase in the CellCycleScoring function embedded in Seurat.

## Trajectory analyses

To determine the potential development lineages of T cell subclusters, we converted the previous Seurat object into Monocle3 (v1.0.0, R package) (*Cao et al., 2019*) object and inferred the trajectory of T cell subclusters at its proper position in pseudotime. Besides, to visualize the major non-linear components of variation across cells, we applied destiny package (v3.1.1, R package) (*Angerer et al., 2016*) to perform the 3D diffusion maps to compute the diffusion components of each cell type.

## TCR clonotype analysis

Cell Ranger VDJ pipeline (v6.1.1, 10× Genomics) was used to process the raw TCR sequence data with default augments and align them to the Ensembl GRCh38 reference (https://cf.10xgenomics.com/supp/cell-vdj/refdata-cellranger-vdj-GRCh38-alts-ensembl-5.0.0.tar.gz). We performed scRepertoire (v1.3.2, R package) (*Borcherding et al., 2020*) to integrate the TCR sequence data with mRNA expression data and used absolute frequency of V(D)J genes to define clonotype groups. The total frequency assigned for different extents of clonal expansion was categorized as follows: hyperexpanded ($20 < X \leq 200$), multiple ($2 < X \leq 20$), double ($1 < X \leq 2$), single ($0 < X \leq 1$).

## Ligand–receptor interactions

To understand communications among tumor cell clusters, we applied CellChat (v1.1.3, R package) (*Jin et al., 2021*) to identify the cell–cell signaling links, inferred the cellular communication network and visualized the major ligand–receptor interaction between each cell cluster.

## Library construction of ST

Representative HNSCC tumor samples were collected for the spatial transcriptomic sequencing. Samples were cut into 6.5×6.5 mm² pieces, embedded in optimal cutting compound media and quickly frozen on dry ice. The frozen tissues were cryosectioned at 10 μm thickness by using the Thermo Scientific CryoStar NX50 cryostat and were placed in the capture area frames on the 10× Visium Spatial slides. Each sample slide was stained with H&E (Hematoxylin Dako #S3309, Eosin, Dako #CS701, bluing buffer #CS702) and the brightfield images were captured via Leica whole-slide scanner at 10× resolution.

Following tissue permeabilization, reverse transcription and cDNA amplification were processed by using Reagent Kit (10× Genomics, #PN-1000184, PN-1000193). Visum spatial libraries were constructed using Visum Spatial Library Construction kit (10× Genomics, #PN-1000184) according to the manufacturer's protocols. Finally, the libraries were sequenced using the Illumina NovaSeq6000 at least 100,000 reads per spot via pair-end 150 bp (PE150) reading strategy (performed by CapitalBio Technology, Beijing).

## Functional scoring and visualization of ST data

We performed alignment, filtering, barcode counting, and UMI counting by the Spaceranger count (v1.3.0) to generate feature-barcode matrix. We performed normalization, high-variance features detection (top 2000 genes), dimensionality reduction, and clusters identification (resolution = 1.0) for the spatially barcoded gene expression data via the standard Seurat pipeline. The P-Tex, Tex, and APC scoring algorithms of ST were similar to the module scoring algorithm of scRNA transcriptome (AddModuleScore) and the gene list of each module was listed in *Supplementary file 5*. Co-localization of P-Tex, Tex, and APC scores was verified by cor.test (stats, v3.6.2, R package) (*Maurage et al., 2013*).

## Multiplex immunohistochemistry

Ten FFPE tissue of HNSCC tumors were sectioned to 4 µm thick for the subsequent multiplex immunohistochemistry via the OPAL Polaris system (Akoya Biosciences). After deparaffinization and hydration, the FFPE slides were manually stained with the CD8 (clone C8/144B, CST, #70306S), PD-1 (clone D7D5W, CST, #84,651T), anti-Ki67 (clone SP6, Abcam, #ab16667), CDK4 (clone D9G3E, CST, #12790), CD24 (10600-1-AP, Proteintech, #10600-1-AP), Anti-EPCAM (clone EPR20532-225, Abcam, #ab223582), and Anti-HLA-DR (clone EPR3692, Abcam, #ab92511) antibodies. The sections were counterstained with spectral DAPI (Akoya Biosciences). The stained slides were imaged and scanned using the Vectra Polaris multispectral imaging system.

## Cell staining strategies for flow cytometry

Single-cell suspensions (100 µL) of HNSCC tumor tissues were stained with CD3 (BD Pharmingen, #555332), CD8 (CST, #300908), PD1 (Biolegend, #329920), UBE2C (Santacruz, #Sc271050), and KLRB1 (Biolegend, #339917) antibodies at 4°C for 30 min under dark conditions. And 7-aminoactinomycin D (7AAD) was used for live/dead discrimination. P-Tex cells were defined as 7AAD-CD3$^+$CD8$^+$PD1$^+$UBE2C$^+$ cells and T-Tex cells were defined as 7AAD-CD3$^+$CD8$^+$PD1$^+$KLRB1$^+$UBE2C$^-$ cells.

## In vivo cell function assays

### Cell culture and proliferation assay

The sorted P-Tex cells and T-Tex cells were cultured in RPMI media containing 10% FBS, penicillin, streptomycin, and 20 IU/mL IL-2, and stimulated with T Cell TransAct (diluted at 1:100, T Cell TransAct, human, Miltenyi Biotec, #130-111-160), with fresh medium replaced every 3 days. After 14 consecutive days, proliferation rate of cells was assessed by flow cytometry.

### Survival assay

The sorted P-Tex and T-Tex cells were cultured in the RPMI media containing 10% FBS, human IL-2 (20 IU/mL), penicillin, and streptomycin in 96-well plates (15,000 cells/well). At the 5th, 10th, and 15th day of growth, cell counting was performed by flow cytometry. Cell viability was determined by propidium iodide (PI) staining on the 15th and 20th day of cell growth. And on the 9th and 16th day, P-Tex and T-Tex cells were stained by UBE2C (Santacruz, #Sc271050), PD1 (Biolegend, #329920), KLRB1 (Biolegend, #339917), and the protein expression was detected by flow cytometry. Software FlowJo was used for data analysis.

### CDK4/6 inhibition test

The sorted P-Tex and T-Tex cells (50,000 cells/well) and the cancer cell line (FD-LSC-1, 4000 cells/well, donated by State Key Laboratory of Biotherapy, West China Medical School, Sichuan University) (*Liu et al., 2020a*) were transferred into 96-well plates, treated with Abemaciclib in gradient

concentrations (0, 10, 100, and 500 nM) for 24, 48, and 72 hr, respectively. Cell proliferation was detected by Cell Counting Kit-8 (CCK-8) according to the manufacturer's instruction. FD-LSC-1 cell line was authenticated by STR profiling. Cells were tested regularly for *Mycoplasma* contamination and none tested positive throughout the studies.

## Survival analysis

We further analyzed the transcriptome data of 500 HNSCC tumor samples (HPV negative: n=410; HPV positive: n=90) in TCGA cohort (*Liu et al., 2020b*). Survival analysis related to gene expression level and functional module score in different HPV status was conducted through the Tumor Immune Estimation Resource (TIMER; cistrome.shinyapps.io/timer) (*Li et al., 2017*). Besides, to predict the proportion of P-Tex, Teff, and T-Tex cells in HPV-positive and HPV-negative HNSCC TCGA cohort, we used CIBERSORT software to deconvolve our scRNA-seq data (534 specific marker genes, *Supplementary file 3*) into the TCGA bulk transcription data for clustering.

## Statistical analysis

Statistical analysis was performed using R (v3.6.3). Wilcoxon rank-sum tests and Chi-square tests were used to compare variables. The hazard ratio and survival curves were estimated via a Cox regression model. A statistical significance was considered at $p < 0.05$.

## Acknowledgements

We would like to thank the staff and students in the Department of Oto-Rhino-Laryngology, West China Biomedical Big Data Center, and Research Core Facility of West China Hospital for giving us kind support of sample collection and experiments. National Natural Youth Science Foundation of China grant 82002868 (RJJ). National Natural Youth Science Foundation of China grant 32100927 (YHP). China Postdoctoral Science Foundation grant 2020M673250 (RJJ). The Science and Technology Department of Sichuan Province grant 2022YFS0066 (ZYB). The Science and Technology Department of Sichuan Province grant 2020YFS0111 (RJJ). The Science and Technology Department of Sichuan Province grant 2021YFS0158 (YHP). West China Hospital, Sichuan University grant ZYJC21027 (ZY). West China Hospital, Sichuan University grant 2019HXBH079 (RJJ). Sichuan University grant 2020SCU12049 (RJJ). The Health Department of Sichuan Province grant 20PJ030 (RJJ).

## Additional information

### Funding

| Funder | Grant reference number | Author |
| --- | --- | --- |
| National Youth Foundation of China | 82002868 | Jianjun Ren |
| National Youth Foundation of China | 32100927 | Haopeng Yu |
| China Postdoctoral Science Foundation | 2020M673250 | Jianjun Ren |
| Sichuan Province Science and Technology Support Program | 2022YFS0066 | Yongbo Zheng |
| Sichuan Province Science and Technology Support Program | 2020YFS0111 | Jianjun Ren |
| Sichuan Province Science and Technology Support Program | 2021YFS0158 | Haopeng Yu |
| West China Hospital, Sichuan University | ZYJC21027 | Yu Zhao |

| Funder | Grant reference number | Author |
| --- | --- | --- |
| West China Hospital, Sichuan University | 2019HXBH079 | Jianjun Ren |
| Sichuan University | 2020SCU12049 | Jianjun Ren |
| Health Department of Sichuan Province | 20PJ030 | Jianjun Ren |

The funders had no role in study design, data collection and interpretation, or the decision to submit the work for publication.

## Author contributions

Danni Cheng, Yufang Rao, Validation, Investigation, Methodology, Writing – original draft, Project administration; Ke Qiu, Minzi Mao, Investigation, Methodology, Writing – original draft, Project administration; Li Li, Yan Wang, Jun Liu, Haiyang Wang, Daibo Li, Lin Yang, Resources, Investigation, Methodology; Yao Song, Junren Chen, Investigation, Methodology; Xiaowei Yi, Conceptualization, Software, Formal analysis, Supervision, Funding acquisition, Investigation, Project administration, Writing – review and editing; Xiuli Shao, Software, Formal analysis; Shao Hui Huang, Software, Formal analysis, Supervision; Yi Zhang, Formal analysis, Supervision, Methodology; Xuemei Chen, Resources, Formal analysis, Methodology; Sisi Wu, Resources, Data curation, Methodology; Shuaishuai Yu, Data curation, Investigation, Methodology; Xingchen Peng, Resources, Investigation; Li Chen, Resources, Validation, Investigation, Methodology; Zhiye Ying, Data curation, Software, Validation, Methodology; Yongbo Zheng, Data curation, Software, Funding acquisition, Validation; Meijun Zheng, Funding acquisition, Validation, Investigation, Visualization; Binwu Ying, Xiaoxi Zeng, Validation, Investigation, Visualization; Wei Zhang, Supervision, Writing – review and editing; Wei Xu, Data curation, Investigation, Visualization, Methodology; Geoffrey Liu, Data curation, Validation, Methodology; Fei Chen, Resources, Data curation, Supervision, Funding acquisition, Validation; Haopeng Yu, Resources, Data curation, Software, Supervision, Funding acquisition; Yu Zhao, Data curation, Software, Supervision, Funding acquisition, Project administration, Writing – review and editing; Jianjun Ren, Conceptualization, Supervision, Funding acquisition, Investigation, Project administration, Writing – review and editing

## Author ORCIDs

Danni Cheng http://orcid.org/0000-0003-0368-6258
Minzi Mao http://orcid.org/0000-0002-8777-6770
Yu Zhao http://orcid.org/0000-0003-1874-908X
Jianjun Ren http://orcid.org/0000-0002-5938-688X

## Ethics

This study was conducted in accordance with the Declaration of Helsinki (as revised in 2013) and was approved by the Biomedical Research Ethics Committee of West China Hospital (2021-908), with the individual consent for each participant.

## Decision letter and Author response

Decision letter https://doi.org/10.7554/eLife.82705.sa1
Author response https://doi.org/10.7554/eLife.82705.sa2

# Additional files

## Supplementary files

- Supplementary file 1. Patients' information.

- Supplementary file 2. Cell numbers of single T cells from 14 head and neck squamous cell carcinoma (HNSCC) samples by single-cell RNA sequencing (scRNA-seq).

- Supplementary file 3. Marker genes of different cell clusters applied in single-cell RNA sequencing (scRNA-seq).

- Supplementary file 4. Transcriptional regulators of top expressed genes in each T cell clusters.

- Supplementary file 5. Functional cell scores of each cell cluster.

- Supplementary file 6. Numbers of shared clonotypes among each cell clusters.
- Supplementary file 7. The overlap coefficients among each cluster.
- Supplementary file 8. Cell numbers of different head and neck squamous cell carcinoma (HNSCC) tumor microenvironment (TME) subclusters by single-cell RNA sequencing (scRNA-seq).
- Supplementary file 9. The proportion of P-Texs, T-Tex, and Teff clusters in HPV[+] and HPV[-] head and neck squamous cell carcinoma (HNSCC) samples in The Cancer Genome Atlas (TCGA) database.
- Supplementary file 10. The communication network of signaling pathways among different cell clusters derived by ligand–receptor interactions.
- Supplementary file 11. The P-Tex, Tex, and APC scores in spatial transcriptomic.
- MDAR checklist

### Data availability

The raw sequence data reported in this paper have been deposited in the Genome Sequence Archive in National Genomics Data Center, China National Center for Bioinformation/Beijing Institute of Genomics, Chinese Academy of Sciences (GSA-Human: HRA003921) that are publicly accessible at https://ngdc.cncb.ac.cn/gsa-human.

The following dataset was generated:

| Author(s) | Year | Dataset title | Dataset URL | Database and Identifier |
|---|---|---|---|---|
| Cheng D, Qiu K, Rao Y, Mao M, Zhao Y, Ren J | 2023 | Single-cell Sequence Data of Human Papillomavirus Positive Head and Neck Squamous Cell Carcinoma | https://ngdc.cncb. ac.cn/gsa-human/ browse/HRA003921 | Genome Sequence Archive in National Genomics Data Center, China National Center for Bioinformation/ Beijing Institute of Genomics, Chinese Academy of Sciences, HRA003921 |

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
