## [Editor Report]

This study provides fundamental insight into the functional impact of CDK4 inhibition on cells in the tumor microenvironment, which is of high importance and interest to the field. The compelling conclusion that proliferative exhausted T cells are associated with response in HPV+ head and neck cancer is supported by the cohort of 14 patients with paired tumor and adjacent normal tissue and rigorous bioinformatic analysis of nearly 50,000 single CD3+ T cell transcriptomes. This work will be of interest to researchers across tumor types and in other immunological fields of study.

---

## [Decision Letter]

**Decision letter after peer review:**

Thank you for submitting your article "Proliferative Exhausted CD8 T Cells Exacerbate Long-lasting Anti-tumor Effects in Human Papillomavirus Positive Head and Neck Squamous Cell Carcinoma" for consideration by eLife. Your article has been reviewed by 1 peer reviewer, and the evaluation has been overseen by a Reviewing Editor and Tadatsugu Taniguchi as the Senior Editor. The reviewer has opted to remain anonymous.

The Reviewing Editor has drafted this to help you prepare a revised submission.

Essential revisions:

1) Temper some claims as indicated in the reviewer comments, ensuring that the conclusions are supported by the data presented.

2) Clarify the use of TCRseq to ensure TCR-based conclusions are biologically relevant.

---

## [Author Response]

Essential revisions:1) Temper some claims as indicated in the reviewer comments, ensuring that the conclusions are supported by the data presented.2) Clarify the use of TCRseq to ensure TCR-based conclusions are biologically relevant.

With the kind help from you, the paper has been carefully revised and all the comments are fully dealt with. Specifically, we have tempered some claims as indicated in the reviewers’ comments, ensuring that the conclusions are supported by the data presented. And we especially clarified the use of TCRseq to ensure TCR-based conclusions are biologically relevant.